# Estimation of Entropy in Constant Space with Improved Sample Complexity

**Maryam Aliakbarpour** [*]     Andrew McGregor [†]     Jelani Nelson [‡]     Erik Waingarten [§]

## Abstract

Recent work of Acharya et al. (NeurIPS 2019) showed how to estimate the entropy of a distribution $\mathcal{D}$ over an alphabet of size $k$ up to $\pm\epsilon$ additive error by streaming over $(k/\epsilon^3) \cdot \mathrm{polylog}(1/\epsilon)$ i.i.d. samples and using only $O(1)$ words of memory. In this work, we give a new constant memory scheme that reduces the sample complexity to $(k/\epsilon^2) \cdot \mathrm{polylog}(1/\epsilon)$. We conjecture that this is optimal up to $\mathrm{polylog}(1/\epsilon)$ factors.

## 1 Introduction

In the field of *streaming algorithms*, an algorithm makes one pass (or few passes) over a database while using memory sublinear in the data it sees to then answer queries along the way or at the data stream's end. Researchers have developed various algorithms, as well as memory lower bounds, for such problems for over four decades [MP80, MG82, AMS99]. For the vast majority of research in the field, the database is assumed to be fixed, and algorithms are then analyzed through the lens of worst case analysis.

In this work, we look to further develop the relationship between streaming algorithms and statistics, specifically studying statistical inference through low-memory streaming algorithms. In this setup, rather than processing a worst-case instance of some fixed database, our input is instead a *distribution* $\mathcal{D}$, and our algorithm processes i.i.d. samples from $\mathcal{D}$ with the goal of inferring its properties. Natural questions then arise, such as understanding the tradeoffs between sample complexity, memory, accuracy, and confidence, or even understanding whether a low-memory algorithm exists at all for a particular inference problem even if we allow the streaming algorithm to draw an unlimited number of samples. Work on streaming algorithms for statistical inference problems began in [GM07], which studied nonparameteric distribution learning, followed by the work of [CLM10], studying low-memory streaming algorithms for use in robust statistics and distribution property testing. Interest in the area later exploded off after work of [SVW16], which explicitly raised the question of whether low memory might place fundamental limits on learning rates, with a flurry of works proving such limitations in response [CMVW16, MM17, KRT17, Raz17, GRT18, SSV19, GRT19, GKR20, GKLR21], starting with a work of [Raz19] on memory/sample tradeoff lower bounds for learning parities ($\mathcal{D}$ generates $(x, \langle w, x \rangle)$ for $x$ uniform in the hypercube with $w$ an unknown parameter, and the goal is to learn $w$).

In this work, following [ABIS19], we focus specifically on the problem of estimating the entropy of an unknown distribution $\mathcal{D}$ over $\{1, \ldots, k\}$ using a low-memory streaming algorithm over i.i.d.

---

[*]Boston University and Northeastern University, Supported by NSF awards CNS-2120667, CNS-2120603, CCF 1934846, and BU's Hariri Institute for Computing; `maryam.aliakbarpour@gmail.com`.

[†]University of Massachusetts Amherst, `mcgregor@cs.umass.edu`. Supported by NSF awards CCF-1934846, CCF-1908849, and CCF-1637536.

[‡]UC Berkeley, `minilek@berkeley.edu`. Supported by NSF award CCF-1951384, ONR grant N00014-18-1-2562, ONR DORECG award N00014-17-1-2127, and a Google Faculty Research Award.

[§]Stanford University, `eaw@cs.columbia.edu`. Part of this work is supported by the National Science Foundation under Award no. 2002201 and Moses Charikar's Simons Investigator Award.

36th Conference on Neural Information Processing Systems (NeurIPS 2022).

samples. It is known that to estimate the entropy up to $\epsilon$ additive error with large constant success probability, without memory constraints the optimal sample complexity is

$$n = \Theta\left(\max\left\{\frac{1}{\epsilon}\frac{k}{\log(k/\epsilon)}, \frac{\log^2 k}{\epsilon^2}\right\}\right)$$

[VV17, VV11, JVHW15, WY16]. Prior work by [BDKR05] also shows that a sublinear number of samples is possible for multiplicative approximation of entropy for distributions whose entropy is sufficiently large. The known optimal algorithms from prior work, however, must remember all samples and hence use $\Omega(n)$ words of memory[5]. The algorithm of [ABIS19] uses only $O(1)$ words of memory, though at the cost of requiring an increased sample complexity of $k \cdot \tilde{O}(1/\epsilon^3)$[6]. In this work, our goal is to address the question: *to what extent was the worsening of sample complexity in previous work necessary to achieve constant memory?*

**Our Contribution.**   We show that using $O(1)$ words of memory[7], it is possible to obtain a sample complexity of $k \cdot \tilde{O}(1/\epsilon^2)$, which is an improvement over the previous memory-efficient sample complexity bound which had cubic dependence on $1/\epsilon$. The starting point of our algorithm revisits a simple estimator proposed by [ABIS19]. Their simple estimator uses $O(k\log^2(k/\epsilon)/\epsilon^3)$ samples to estimate the entropy in constant space. Our novel contribution is a modification which estimates a bias incurred by the estimator; this change allows us to use only $O(k\log^2 k\log^2(1/\epsilon)/\epsilon^2)$ samples. With the simple estimator with improved sampled complexity in hand, we show how an "interval-based" algorithm, similar to the one in [ABIS19], improves the dependence on $k$ to $k \cdot \tilde{O}(1/\epsilon^2)$.

We remark that there has been other work on estimating entropy in the data streaming model [BG06, CCM10, HNO08, KNW10, JW19], but those works are qualitatively different from our own current work and that of [ABIS19]. Specifically, they take the worst case point of view, where the stream items are not drawn i.i.d. from a distribution, but rather the stream itself is viewed as a worst-case input and the goal is to estimate its empirical entropy. In that model, $O(1)$ memory algorithms for $\pm\epsilon$ additive estimation to entropy provably do not exist, as there is a known memory lower bound of $\Omega(1/\epsilon^2)$ bits [JW19].

**Overview of Approach.**   We start by describing the basic algorithm of [ABIS19]. Their basic estimator takes a single random sample $\boldsymbol{i} \sim \mathcal{D}$, followed by $N$ more i.i.d. samples. Then, they define $\boldsymbol{N}_x$ to be the number of these $N$ samples equal to $\boldsymbol{i}$. The estimate $\hat{p}_{\boldsymbol{i}} := \boldsymbol{N}_x/N$ is an unbiased estimator of of the probability $p_{\boldsymbol{i}}$ of $\boldsymbol{i}$ according to $\mathcal{D}$, and for large $N$, $\log(1/\hat{p}_{\boldsymbol{i}})$ is a reasonable estimator for the entropy $H = H(\mathcal{D}) = \mathbf{E}\left[\log(1/p_{\boldsymbol{i}})\right]$ of $\mathcal{D}$. One can then average many such independent estimates. There is an additional technical detail, that $\hat{p}_{\boldsymbol{i}}$ may be zero (if $\boldsymbol{N}_x$ is zero), which is fixed via a "one-smoothing" trick of actually setting $\hat{p}_{\boldsymbol{i}} := (\boldsymbol{N}_x + 1)/N$ (which introduces an acceptably small amount of additional bias when $N$ is sufficiently large).

Our improvement begins with the observation that $\log(1/\hat{p}_{\boldsymbol{i}})$ is *not* an unbiased estimator for $H$. We first propose a similar but different estimator to the previous simple estimator. We also begin by taking a random sample $\boldsymbol{i} \sim \mathcal{D}$; however, rather than letting $\boldsymbol{N}_x$ be sampled from the binomial distribution $\mathsf{Bin}(N, p_{\boldsymbol{i}})$, we sample a negative binomial random variable $\boldsymbol{X}$, which is the number of additional draws to see $i$ exactly $t$ more times ($t$ is a parameter of the algorithm). Henceforth we let $\mathrm{NB}(t, p)$ denote such a negative binomial random variable, where the underlying Bernoulli experiment has success probability $p$. Then $\mathbf{E}[\boldsymbol{X}] = t/p_{\boldsymbol{i}}$, and we will use $\log(\boldsymbol{X}/t)$ as a reasonable estimate of $\log(1/p_{\boldsymbol{i}})$. This estimator is also biased, but we can correct for this bias using a few more samples.

Specifically, let $\boldsymbol{Y} = \boldsymbol{X}p_{\boldsymbol{i}}/t$ and consider the degree-$r$ Taylor expansion of our estimate $\log(\boldsymbol{X}/t)$ and the ideal quantity $\log(1/p_{\boldsymbol{i}})$. As it will turn out, the expectation of the degree-$r$ Taylor expansion of $\log(\boldsymbol{X}/t) - \log(1/p_{\boldsymbol{i}}) = \log \boldsymbol{Y}$ is a degree-$r$ polynomial in $p_{\boldsymbol{i}}$. By drawing $r$ additional samples,

---

[5]As in prior work, we use a "word", or "machine word", to denote a unit of memory that can hold $\Theta(\log(k/\epsilon))$ bits. Essentially, a machine word is large enough to hold the name of an item in the alphabet, as well as the value of $\epsilon$.

[6]We use $\tilde{O}(f)$ to denote a function which is $O(f \cdot \mathrm{poly}(\log f))$

[7]More precisely, we provide a uniform algorithm which given any $k, \epsilon$ generates a program with source code of size $O(\log\log(1/\epsilon))$ words, and that fixed program can then process any stream in $O(1)$ words of working memory.

we may design an estimator for this polynomial, and subtract it from $\log(\boldsymbol{X}/t)$. Correcting some of the bias in this way gives us our improved estimate for $\log(1/p_{\boldsymbol{i}})$. Our analysis of this scheme shows that a sample complexity of $(k/\epsilon^2) \cdot \mathrm{polylog}(k/\epsilon)$ suffices. We then describe and analyze an improved algorithm in Section 3, which achieves $(k/\epsilon^2) \cdot \mathrm{polylog}(1/\epsilon)$ sample complexity by additionally incorporating a "bucketing" scheme, similar to one proposed in [ABIS19]. The idea is to partition the possibilities for values of $\boldsymbol{X}$ into disjoint intervals $I_\ell = [b_{\ell-1}, b_\ell)$ for $\ell = 1, 2, \ldots, L$ and optimized choices of the breakpoints $b_\ell$, then estimate both $\mathbf{Pr}[\boldsymbol{X} \in I_\ell]$ and the conditional contributions to entropy conditioned on $\boldsymbol{X} \in I_\ell$ for each $\ell$. By estimating separately for each $I_\ell$, one can show that the conditional variance is reduced to obtain an overall smaller sample complexity of $(k/\epsilon^2) \cdot \mathrm{polylog}(1/\epsilon)$, a strict improvement over that of [ABIS19]; details are in Section 3.

**Lower bounds:**   Diakonikolas et al. [DGKR19] show lower bounds for sample-memory tradeoffs for testing uniformity of distributions. They construct a distribution $p$ over $[2k]$ such that for every $i \in [k]$, the probabilities of the elements $2i$ and $2i - 1$ are $(1 + \sqrt{\epsilon})/(2k)$ and $(1 - \sqrt{\epsilon})/(2k)$ while the order is picked randomly. They show that any streaming algorithm that uses $m$ bits of memory, $n$ samples, and can distinguish $p$ from the uniform distribution over $[2k]$ with high constant probability requires: (i) $m \cdot n = \Omega(\frac{k}{\epsilon})$, and (ii) if $n \leq k^{0.9}$ and $m \geq n^2/k^{0.9}$, $m \cdot n = \Omega(\frac{k \log k}{\epsilon^2})$. It is not hard to see that the entropy of the difference between uniform distribution and $p$ is $\Theta(\epsilon)$. While this trade-off indicates a more general relation between the memory usage and samples, only (i) applies to constant memory algorithms (since (ii) requires $n \leq k^{0.9}$). With a word being $\log(k/\epsilon)$ bits, (i) leads to a lower bound of $\Omega(k/(\epsilon \log(k/\epsilon)))$ samples (the same as with unbounded memory). This leaves open whether the sample complexity for $O(1)$-word algorithms is $\Omega(k/\epsilon)$ or $\Omega(k/\epsilon^2)$. We speculate the latter, and we discuss our conjecture in Section 4.

## 2   A Simple Algorithm and Analysis

Let $k \in \mathbb{N}$, and $\mathcal{D}$ be an unknown distribution supported on $[k]$. For any $i \in [k]$, we denote the probability that $i \in [k]$ is sampled by $\mathcal{D}$ as $p_i$. The goal is to design a low-space streaming algorithm which receives independent samples from $\mathcal{D}$ and outputs an estimate to the entropy:

$$H(\mathcal{D}) \stackrel{\mathrm{def}}{=} \sum_{i=1}^{k} p_i \log\left(\frac{1}{p_i}\right) = \mathbf{E}_{\boldsymbol{i} \sim \mathcal{D}}\left[\log\left(\frac{1}{p_{\boldsymbol{i}}}\right)\right],$$

where logarithms above and throughout this paper are base-2, unless otherwise stated.

### 2.1   An Estimator for $\log(1/p_{\boldsymbol{i}})$

As mentioned in Section 1, similarly to [ABIS19] the algorithm aims to estimate $H(\mathcal{D})$ by taking a sample $\boldsymbol{i} \sim \mathcal{D}$ and estimating $\log(1/p_{\boldsymbol{i}})$. Then, averaging these estimates will give an estimator for $H(\mathcal{D})$ (albeit with a super-linear dependence on $k$, which we fix in Section 3). We describe the estimator in Figure 1.

There are three main steps in the analysis. In the first, we show that the estimator has small bias. The second is showing that the above estimator has low variance. Finally, we show that the estimator may be computed with few bits. In Figure 1, we set $r = \Theta(\log(1/\epsilon))$ and $t = \Theta(\log^2(1/\epsilon))$ to obtain an estimator whose bias is at most $\epsilon$ and variance is at most $O(\log^2 k)$. It then follows that repeating the estimate of $\log(1/p_{\boldsymbol{i}})$ for $O(\log^2 k/\epsilon^2)$ i.i.d. chosen $\boldsymbol{i} \sim \mathcal{D}$ gives the desired estimate with probability at least $2/3$. These parameter settings establish the following theorem:

**Theorem 1.** *There exists a single-pass data stream algorithm using $O(1)$ words of working memory that processes a stream of $O(k\epsilon^{-2} \log^2 k \log^2(1/\epsilon))$ i.i.d. samples from an unknown distribution $\mathcal{D}$ on $[k]$ and returns an additive $\epsilon$ approximation of $H(\mathcal{D})$ with probability $2/3$.*

The space complexity in the theorem above follows since computing the estimator just requires maintaining integers in the sets $[k], [t]$, and $[r]$, as well as computing a low-degree polynomial. To compute the average of multiple estimators in small space it suffices to compute the sum of the estimates where each estimator is computed in series. The sample complexity bound (given the specified parameters) in the above theorem follows directly from the sample complexity of `LogEstimator`. By virtue of the fact our estimators are based on negative binomial distributions ($\boldsymbol{X}$

Subroutine `LogEstimator`$(\mathcal{D}, i)$

**Input:** Sample access to a distribution $\mathcal{D}$ supported on $[k]$, an index $i \in [k]$ where $p_i \neq 0$.
**Output:** A number $\boldsymbol{\eta} \in \mathbb{R}_{\geq 0}$, which is our bias estimate.

1. We draw enough samples from $\mathcal{D}$ so that $i$ is sampled exactly $t$ times, and let $\boldsymbol{X} \in \mathbb{N}$ denote the number of samples taken.

2. For $r \in \mathbb{N}$, let $f \colon \mathbb{R} \to \mathbb{R}$ denote the degree-$r$ Taylor expansion of $\log z$ centered at 1, and $h_t \colon [0,1] \to \mathbb{R}$ be the degree-$r$ polynomial satisfying

$$h_t(\rho) = \mathbf{E}_{\boldsymbol{Z} \sim \mathrm{NB}(t,\rho)}\left[ f\left( \frac{\boldsymbol{Z} \cdot \rho}{t} \right) \right].$$

Finally, $g \colon [0,1]^r \to \mathbb{R}$ is the linear function with $g(\rho, \rho^2, \dots, \rho^r) = h_t(\rho)$. We take $r$ additional independent samples from $\mathcal{D}$, and for $j \in [r]$, we let $\boldsymbol{B}_j$ be the indicator random variable that the first $j$ samples were all $i$. Note that $\{\boldsymbol{B}_j\}_{j \in [r]}$ can be encoded using a single counter requiring $\log r$ bits.

3. We return

$$\boldsymbol{\eta} \stackrel{\text{def}}{=} \log\left( \frac{\boldsymbol{X}}{t} \right) - g\left( \boldsymbol{B}_1, \boldsymbol{B}_2, \dots, \boldsymbol{B}_r \right).$$

Figure 1: Description of the estimator for $\log(1/p_i)$.

in Figure 1 is the number of Bernoulli trials until $t$ successes), this in turn follows directly from the expectation of negative binomial distributions:

**Fact 2.1** (Expected Sample Complexity of `LogEstimator`). *Suppose we draw $\boldsymbol{i} \sim \mathcal{D}$ and execute* `LogEstimator`$(\mathcal{D}, \boldsymbol{i})$. *Then, the expected sample complexity is*

$$\sum_{i=1}^{k} p_i \left( r + \frac{t}{p_i} \right) = r + tk.$$

Although the number of samples we draw is a random variable that is only bounded in expectation, note that it implies the existence of a good algorithm that always has a bounded sample complexity: namely, we can simply terminate the algorithm early and output Fail if it draws a large constant factor times more samples than we expect, which happens with low probability by Markov's inequality.

Before moving on to the showing the properties of the estimator, we verify that $h_t(\rho)$ is a degree-$r$ polynomial.

**Lemma 2.2.** *For any $r \in \mathbb{N}$, let $f \colon \mathbb{R} \to \mathbb{R}$ denote the degree-$r$ Taylor expansion of $\log(z)$ centered at 1. Then, for any $\rho > 0$ and $t \in \mathbb{N}$,*

$$h_t(\rho) = \mathbf{E}_{\boldsymbol{Z} \sim NB(t,\rho)}\left[ f\left( \frac{\boldsymbol{Z} \cdot \rho}{t} \right) \right]$$

*is a polynomial of degree at most $r$.*

*Proof.* Recall that the random variable $\boldsymbol{Z} \sim \mathrm{NB}(t,\rho)$ is the number of independent trials from a $\mathrm{Ber}(\rho)$ distribution before one sees $t$ successes. Furthermore, $f$ is the degree-$r$ Taylor expansion of $\log z$ centered at 1, and

$$f(z) = \frac{1}{\ln(2)} \sum_{i=1}^{r} \frac{(-1)^{i+1}}{i} \cdot (z-1)^r.$$

By linearity of expectation, it suffices to show that for every $j \in \{1, \dots, r\}$, $\mathbf{E}_{\boldsymbol{Z}}\left[ (\boldsymbol{Z}\rho/t - 1)^j \right]$ is a degree-$j$ polynomial in $\rho$. Note that $\boldsymbol{Z}$ is a sum of $t$ independent $\mathrm{Geo}(\rho)$ random variables, so by expanding $(\frac{1}{t}\sum_{i=1}^{t} \boldsymbol{G}_i\rho - 1)^j$ and applying linearity of expectation once more, it suffices to show that

$$\mathbf{E}_{\boldsymbol{G} \sim \mathrm{Geo}(\rho)}\left[ (\boldsymbol{G} \cdot \rho)^j \right] = \rho^j \mathbf{E}_{\boldsymbol{G} \sim \mathrm{Geo}(\rho)}\left[ \boldsymbol{G}^j \right] = \rho^j \sum_{k=1}^{\infty} \rho(1-\rho)^{k-1} k^j$$

is a degree-$j$ polynomial in $\rho$. We note that this latter term, $\mathbf{E}_G[G^j]$ may be expressed as $\rho \cdot \text{Li}_{-j}(1-\rho)$, where $\text{Li}_{-j}(\cdot)$ is the polylogarithm function (see [Wei]). $\text{Li}_{-j}(1-\rho)$ happens to be a rational function, where the denominator is exactly $\rho^{j+1}$, which cancels the $\rho^{j+1}$ term. In addition, the numerator of $\text{Li}_{-j}(1-\rho)$ is a degree-$j$ polynomial in $\rho$, which gives the desired polynomial representation. $\qquad\square$

Finally, it will be useful for the variance calculation to show that the correction term is always bounded, which we show here.

**Lemma 2.3.** *There exists a universal constant $c > 0$ such that, for any $r, t \in \mathbb{N}$, if we let $g\colon [0,1]^r \to \mathbb{R}$ be the linear function where $g(\rho, \rho^2, \ldots, \rho^r) = h_t(\rho)$, then $g(b) \in [-c, c]$ for all $b \in \{0,1\}^r$.*

*Proof.* Recall $g\colon [0,1]^r \to \mathbb{R}$ is the linear function where $g(\rho, \rho^2, \ldots, \rho^r) = h_t(\rho)$. Hence, in order to show that $g\colon \{0,1\}^r \to \mathbb{R}$ is bounded, it suffices to show that the sum-of-magnitudes of the $r + 1$ coefficients of $h_t$ is bounded. Since we have

$$h_t(\rho) = \mathbf{E}_{\mathbf{Z}\sim\text{NB}(t,\rho)}\left[f\left(\frac{\mathbf{Z}\cdot\rho}{t}\right)\right] = \frac{1}{\ln(2)}\sum_{i=1}^{r}\frac{(-1)^{i+1}}{i}\cdot\mathbf{E}_{\mathbf{Z}\sim\text{NB}(t,\rho)}\left[\left(\frac{\mathbf{Z}\cdot\rho}{t}-1\right)^i\right].$$

Notice that in Lemma 2.2, we showed that each $\mathbf{E}_{\mathbf{Z}}[(\mathbf{Z}\rho/t - 1)^i]$ is a degree-$i$ polynomial in $\rho$, and the bound (3) implies that, for each $i \in \{1, \ldots, r\}$ these polynomials are at most $(O(i/\sqrt{t}))^i$ in magnitude. Furthermore, since these are degree-$i$ polynomials bounded in $[0,1]$, we conclude (by Lemma 4.1 in [She13]), that the coefficients in $\mathbf{E}_{\mathbf{Z}}[(\mathbf{Z}\rho/t - 1)^i]$ are at most $(O(i/\sqrt{t}))^i$. In particular, we have that the $r$ coefficients of $h_t(\rho)$ are at most

$$\sum_{i=1}^{r}\frac{1}{i}\cdot\left(O(i/\sqrt{t})\right)^i \leq \sum_{i=1}^{r}\left(O(i/\sqrt{t})\right)^i = O(1/\sqrt{t})$$

because $r/\sqrt{t}$ can be made an arbitrarily small constant. To show that $g\colon \{0,1\}^r \to \mathbb{R}$ is bounded, we add the magnitudes of the $r$ coefficients, which is $O(r/\sqrt{t}) = O(1)$ when $r = O(\log(1/\epsilon))$ and $t = O(\log^2(1/\epsilon))$. $\qquad\square$

## 2.2 Bounding Bias of Estimator

**Lemma 2.4.** *Let $\mathcal{D}$ be any distribution and consider any $i \in [k]$. If, for $\epsilon \in (0,1)$, we instantiate* `LogEstimator`$(\mathcal{D}, i)$ *with $r = \Theta(\log(1/\epsilon))$ and $t = \Theta(\log^2(1/\epsilon))$, which produces the random variable $\boldsymbol{\eta}$, then $\left|\mathbf{E}[\boldsymbol{\eta}] - \log\left(\frac{1}{p_i}\right)\right| \leq \epsilon.$*

The remainder of the section constitutes the proof of Lemma 2.4, which will follow from a sequence of claims.

**Claim 2.5.** *In an execution of* `LogEstimator`$(\mathcal{D}, i)$*, let $\mathbf{X}$ and $\boldsymbol{\eta}$ be defined as in Line 1 and Line 3 of Figure 1, and let $\mathbf{Y} = \mathbf{X} \cdot p_i/t$. Then,*

$$\mathbf{E}[\boldsymbol{\eta}] - \log\left(\frac{1}{p_i}\right) = \mathbf{E}_{\mathbf{X}}[h(\mathbf{Y})],$$

*where $h(z)$ is the error in the degree-$r$ Taylor expansion of $\log z$ at $1$.*

**Lemma 2.6.** *For any $\epsilon \in (0,1)$, letting $r = \Theta(\log(1/\epsilon)$ and $t = \Theta(\log^2(1/\epsilon))$, we have that for $p_i > 0$, $|\mathbf{E}_{\mathbf{X}}[h(\mathbf{Y})]| \leq \epsilon.$*

*Proof.* Whenever $z \in (0, 2)$, we may write $\log(z)$ as its Taylor expansion centered at $1$. In particular, we have

$$|h(z)| = |\log(z) - f(z)|$$

$$= \left| \frac{1}{\ln(2)} \sum_{\ell=1}^{\infty} \frac{(-1)^{\ell+1}(z-1)^{\ell}}{\ell} - \frac{1}{\ln(2)} \sum_{\ell=1}^{r} \frac{(-1)^{\ell+1}(z-1)^{\ell}}{\ell} \right|$$

$$= \left| \frac{1}{\ln(2)} \sum_{\ell=r+1}^{\infty} \frac{(-1)^{\ell+1}(z-1)^{\ell}}{\ell} \right|$$

$$= \left| \frac{(z-1)^r}{\ln(2)} \sum_{\ell=1}^{\infty} (-1)^{\ell} \cdot \frac{(z-1)^{\ell}}{r+\ell} \right| = |z-1|^r \left| \frac{1}{\ln(2)} \sum_{\ell=1}^{\infty} (-1)^{\ell} \cdot \frac{(z-1)^{\ell}}{r+\ell} \right| \qquad (1)$$

First, consider the case that $z \in (1, 2)$. Then, we may re-write (1) as

$$|z-1|^r \left| \frac{1}{\ln(2)} \sum_{\ell=1}^{\infty} (-1)^{\ell} \cdot \frac{(z-1)^{\ell}}{r+\ell} \right|$$

$$= |z-1|^r \cdot \left| \frac{1}{\ln(2)} \sum_{\ell=1}^{\infty} \frac{(z-1)^{2\ell-1}}{r+2\ell} \left( \frac{r+2\ell}{r+2\ell-1} - (z-1) \right) \right|. \qquad (2)$$

Whenever $z \in (1, 2)$, then every term in the right-most summation of (2) is positive; indeed, $(z-1)^{2\ell-1}/(r+2\ell) > 0$ because $z > 1$, and $(r+2\ell)/(r+2\ell-1) > 1$ while $(z-1) < 1$. In particular, for $z \in (1, 2)$, we may upper bound the right-most summation in (2) by upper bounding each term. For every $\ell \in \mathbb{N}$, we may upper bound each term

$$\frac{(z-1)^{2\ell-1}}{r+2\ell} \left( \frac{r+2\ell}{r+2\ell-1} - (z-1) \right) \leq \frac{(z-1)^{2\ell-1}}{2\ell} \left( \frac{2\ell}{2\ell-1} - (z-1) \right)$$

$$= \frac{(z-1)^{2\ell-1}}{2\ell-1} - \frac{(z-1)^{2\ell}}{2\ell}.$$

Plugging this upper bound into each term of (2), we have that $z \in (1, 2)$ satisfies

$$|h(z)| \leq |z-1|^r \left| \frac{1}{\ln(2)} \sum_{\ell=1}^{\infty} \left( \frac{(z-1)^{2\ell-1}}{2\ell-1} - \frac{(z-1)^{2\ell}}{2\ell} \right) \right|$$

$$= |z-1|^r \left| \frac{1}{\ln(2)} \sum_{\ell=1}^{\infty} (-1)^{\ell+1} \cdot \frac{(z-1)^{\ell}}{\ell} \right| = |z-1|^r |\log z|.$$

We now consider $z \in (0, 1)$. Here, every term in the right-most summation in (1), $(-1)^{\ell}(z-1)^{\ell}/(r+\ell)$, is positive. So we upper bound

$$|h(z)| = |z-1|^r \left| \frac{1}{\ln(2)} \sum_{\ell=1}^{\infty} (-1)^{\ell} \cdot \frac{(z-1)^{\ell}}{r+\ell} \right|$$

$$\leq |z-1|^r \left| \frac{1}{\ln(2)} \sum_{\ell=1}^{\infty} (-1)^{\ell} \cdot \frac{(z-1)^{\ell}}{\ell} \right|$$

$$= |z-1|^r |\log z|.$$

The final case occurs when $z \geq 2$, and we may no longer use the series representation. However, in this case, we have

$$|h(z)| \leq |\log z| + |f(z)| \leq |\log z| + \sum_{\ell=1}^{r} \frac{|z-1|^{\ell}}{\ell} \leq |\log z| + r|z-1|^{r+1}.$$

In all cases, we have

$$|\mathbf{E}_X[h(Y)]| \leq \mathbf{E}_X[|h(Y)|] \leq O(r) \cdot \mathbf{E}_X\left[|Y-1|^{r+1}\right] + \epsilon/2 + \mathbf{E}_X[\mathbb{1}\{Y \leq 1/10\} \log(1/Y)],$$

where we used the fact that $\boldsymbol{Y} > 0$ and $r = \Theta(\log(1/\epsilon))$ to say $(9/10)^r < \epsilon/2$. In order to bound the above two quantities, we use the fact that the random variable $\boldsymbol{Y}$ is a subgamma random variable and thus has good concentration around its mean (which is 1 for the case of $\boldsymbol{Y}$), giving the desired inequality.

**Definition 2.7** (Subgamma Random Variable). *For $\sigma, B \in \mathbb{R}$, a random variable $\boldsymbol{Z}$ with expectation $\mu$ is $(\sigma, B)$-subgamma if for all $\lambda \in \mathbb{R}$ with $|\lambda| < 1/|B|$,*

$$\psi_{\boldsymbol{Z}}(\lambda) \stackrel{def}{=} \ln\left(\mathbf{E}\left[e^{\lambda(\boldsymbol{Z}-\mu)}\right]\right) \leq \frac{\lambda^2\sigma^2}{2(1-\lambda|B|)}.$$

It is not hard to verify (see Section B) that the random variable $\boldsymbol{Y}$ is centered at 1, and that there are constants $\alpha, \beta \in \mathbb{R}_{\geq 0}$ so $\boldsymbol{Y}$ is $(\alpha/\sqrt{t}, \beta/t)$-subgamma. Then, by taking the Taylor expansion of $\mathbf{E}\left[e^{\lambda(\boldsymbol{Y}-1)}\right]$, we have that for any $|\lambda| < t/\beta$, and any $j \in \mathbb{N}$,

$$\mathbf{E}_{\boldsymbol{X}}\left[|\boldsymbol{Y}-1|^j\right] \leq \frac{j!}{\lambda^j} \cdot \exp\left(\frac{\alpha^2\lambda^2}{2t(1-\lambda\beta/t)}\right) \leq \frac{\alpha^j j!}{t^{j/2}} \cdot e^3, \tag{3}$$

by picking $\lambda = \sqrt{t}/\alpha$, which is less than $t/\beta$ for large enough $t$. Letting $j = r+1$ and setting $t = O(r^2)$, we get the desired bound of $o(\epsilon/r)$. In order to bound $\mathbf{E}_{\boldsymbol{X}}[\mathbb{1}\{\boldsymbol{Y} \leq 1/10\}\log(1/\boldsymbol{Y})]$, we compute it explicitly, and recall that $\boldsymbol{X} \geq t$, so that the above event is satisfied only if $p_i \leq 1/10$.

$$\mathbf{E}_{\boldsymbol{X}}\left[\mathbb{1}\{\boldsymbol{Y} \leq 1/10\}\log(1/\boldsymbol{Y})\right] \leq \mathbf{E}_{\boldsymbol{X}}\left[\frac{\mathbb{1}\{\boldsymbol{Y} \leq 1/10\}}{\boldsymbol{Y}}\right]$$

$$= \sum_{\ell=t}^{t/(10p_i)}\binom{\ell-1}{t-1}p_i^t(1-p_i)^{\ell-t} \cdot \frac{t}{\ell p_i} \leq \frac{t}{10p_i}\max_{\ell\in[t,t/(10p_i)]}\left(\frac{e(\ell-1)}{t-1}\right)^{t-1}p_i^{t-1} \cdot \frac{t}{\ell}$$

$$\leq \frac{t}{10}\max_{\ell\in[t,t/(10p_i)]}\left(\frac{e^2(\ell-1)}{t-1}\right)^{t-2}p_i^{t-2} = \exp(-\Omega(t)). \square$$

# 3 Improving Sample Complexity via Bucketing

In this section, we focus on estimating the expected value of $\log(\boldsymbol{X}/t)$ with error at most $\epsilon$. Our goal here is to remove the $\text{poly}(\log k)$ dependencies in the sample complexity of estimation. In particular, we prove the following theorem, which improves on the dependence of $k$ in Theorem 1.

**Theorem 2.** *There exists a single-pass data stream algorithm using $O(1)$ words of working memory that processes a stream of $O(k\log^4(1/\epsilon)/\epsilon^2)$ i.i.d. samples from an unknown distribution $\mathcal{D}$ on $[k]$ and returns an additive $\epsilon$ approximation of $H(\mathcal{D})$ with probability at least $2/3$.*

Given the work done in Section 2, it will suffice to estimate the quantity $H$ (we give the explicit reduction in Lemma 3.1 shortly):

$$H := \mathbf{E}_{\boldsymbol{i}\sim\mathcal{D}, \boldsymbol{X}\sim\text{NB}(t,p_i)}\left[\log\left(\boldsymbol{X}/t\right)\right], \tag{4}$$

where $t$ is set to $\Theta(\log^2(1/\epsilon))$, such that we can then apply the correction term of Section 2. Recall that the randomness in the above expectation is taken over the random choice of $\boldsymbol{i} \sim \mathcal{D}$, and $\boldsymbol{X}$ is a negative binomial random variable drawn from $\text{NB}(t, p_{\boldsymbol{i}})$. First, we show that it suffices to estimate (4) in order to estimate the entropy, given our tools from Section 2.

**Lemma 3.1.** *Consider a fixed distribution $\mathcal{D}$, and for $\epsilon > 0$ suppose $\hat{H} \in \mathbb{R}$ is such that $|H - \hat{H}| \leq \epsilon$. Then, there exists a $O(1)$ word streaming algorithm which given $\hat{H}$ and using an additional $O(\log(1/\epsilon)/\epsilon^2)$ independent samples from $\mathcal{D}$, outputs an estimate to the entropy of $\mathcal{D}$ up to error $\pm 2\epsilon$ with probability at least $0.9$.*

It thus suffices to design an algorithm to estimate (4). Our approach is to use a bucketing scheme. At a high level, we partition the range of $\boldsymbol{X}$ into $L$ intervals: $I_1, I_2, \ldots, I_L$. We compute the conditional expectation of $\log(\boldsymbol{X}/t)$ in each interval separately. Then, we take the weighted average of these conditional expectations, where the weights are determined by the probability of the intervals.

**Unbounded $X$:**  As specified above, the random variable $X$ is a mixture of negative binomial random variables, so $X$ may be unbounded. In addition, if we had sampled $i \sim \mathcal{D}$ where $p_i$ was very small, $X$'s value will tend to be very large. It will be convenient to introduce a parameter $X_{\max} \in \mathbb{N}$ and consider the random variable $X' := \min(X, X_{\max})$. Let $\tilde{H}$ denotes the expected value of $X'$:

$$\tilde{H} := \mathbf{E}_{i,X}\big[\log(X'/t)\big] \,.$$

For the rest of the section, we will seek to approximate $\tilde{H}$, and the fact that this is a good estimate for $H$ follows from the following lemma.

**Lemma 3.2.** *Let $i \sim \mathcal{D}$, and let $X$ and be a negative binomial random variable from $NB(t, p_i)$. Let $X'$ be the bounded version of $X$: $X' := \min(X, X_{\max})$. Let $t \in \mathbb{N}$ and $\epsilon \in (0, 1)$. If we set $X_{\max} = tk/(\ln(2)\epsilon)$, then*

$$\left|H - \tilde{H}\right| = \mathbf{E}_{i,X}\big[\log(X/t) - \log(X'/t)\big] \le \epsilon \,.$$

**Comparison to related work:**  It is worth noting that the proofs in this section are inspired by the work of [ABIS19]. The authors used a similar bucketing technique to estimate entropy. While the structure of our proof is similar, there are subtle differences between our work and what they did. First, we are focusing on estimating different quantities. In particular, we work with an unbounded random variable while their estimator is bounded. Moreover, they have a two-step bucketing system where they draw a sample $i$ and two estimates for $p_i$; they use one estimate for detecting which bucket falls into and the second one to estimate entropy in that bucket. One of the complications of this approach is that the second estimator may fall into a different bucket; Thus, they have to "clip" the second estimator to make sure it is close to the bucket of the first estimator. We have circumvented these hurdles by using the same estimate for detecting which bucket we are in and estimating $\log(X/t)$ in that bucket.

**The algorithm:**  We write $\tilde{H}$ in terms of conditional expectation in the intervals.

$$\tilde{H} = \sum_{\ell=1}^{L} \underbrace{\mathbf{Pr}_{i\sim\mathcal{D}, X\sim\mathrm{NB}(t,p_i)}\big[X' \in I_\ell\big]}_{q_\ell :=} \cdot \underbrace{\mathbf{E}_{i\sim\mathcal{D}, X\sim\mathrm{NB}(t,p_i)}\big[\log(X'/t) \mid X' \in I_\ell\big]}_{H_\ell :=} \,.$$

Let $q_\ell$ denote the probability of $X'$ being in $I_\ell$, and $H_\ell$ denote the conditional expectation in $I_\ell$. Our algorithm estimate $q_\ell$ and $H_\ell$ for each interval to find an estimate for $\tilde{H}$. Below we give a brief description of our algorithm, and the pseudocode can be found in Algorithm 1.

Below, we define $b_0 = t < b_1 < \cdots < b_L = X_{\max}$ to be $L + 1$ parameters (which we will set shortly) that denote the boundary points of the intervals:

$$I_\ell = [b_{\ell-1}, b_\ell) \qquad \forall i \in [L-1], \qquad I_L = [b_{L-1}, b_L] \,.$$

For each interval $I_\ell$, we draw $r_\ell$ samples from $\mathcal{D}$, namely $i_1, \ldots, i_{r_\ell} \sim \mathcal{D}$. For each $i_j$, we start drawing samples from $\mathcal{D}$ in the process of drawing a negative binomial random variable $X_j \sim \mathrm{NB}(t, p_{i_j})$; then, we will set $X'_j = \min(X_j, X_{\max})$. Furthermore, we will only consider $X'_j$'s that fall in $I_\ell$, which means that we can stop early if we already know $X'_j$ will be too large. In particular, if we draw $b_\ell$ samples and have not observed $t$ instances of $i_j$, we can already conclude $X'_j$ is not in $I_\ell$ and stop sampling. Among these $r_\ell$ samples $\{i_1, \ldots, i_{r_\ell}\}$, let $c_\ell$ denote the number of $X'_j$'s that fall into $I_\ell$. We estimate the weight of each bucket by $\hat{q} := c_\ell/r_\ell$. For the last bucket, we set $\hat{q}_\ell$ in a way that the sum of the weight is one:

$$\hat{q}_\ell = \frac{c_\ell}{r_\ell}, \qquad \forall j = 1, \ldots, L-1, \qquad \hat{q}_L := 1 - \sum_{j=1}^{L-1} \hat{q}_L \,.$$

Also, we compute an average of $\log(X'_j/t)$ of such $X'_j$'s and denote it by $\hat{H}_\ell$:

$$\hat{H}_\ell = \frac{\sum_{j=1}^{r_\ell} \mathbb{1}\{X'_j \in I_\ell\} \cdot \log(X'_j/t)}{c_\ell} \qquad \forall \ell = 1, \ldots, L \,.$$

In these definitions, we take $\hat{H}_L = \log(b_L/t)$ if $c_L = 0$ and for the sake of analysis $\hat{H}_\ell = H_\ell$ if $c_\ell = 0$ for $\ell < L$; note that the value of $\hat{H}_\ell$ will be multiplied by $0$ in this case and hence we may define $\hat{H}_\ell$ arbitrarily. Our estimate for $\tilde{H}$ is the weighted sum of $\hat{H}_\ell$, i.e., $\hat{H} = \sum_{\ell=1}^{L} \hat{q}_\ell \cdot \hat{H}_\ell$.

---

**Algorithm 1** Estimating $\mathbf{E}[\log \boldsymbol{X}/t]$ via Bucketing

---
1: **procedure** LOGESTIMATOR($k$, $\epsilon$, sample access to $\mathcal{D}$)
2:     $\hat{\boldsymbol{H}} \leftarrow 0$
3:     **for** $\ell = 1, 2, \ldots, L$ **do**
4:         $\boldsymbol{c}_\ell \leftarrow 0$, $\hat{\boldsymbol{H}}_\ell \leftarrow 0$
5:         **for** $r_\ell$ times **do**
6:             Draw $\boldsymbol{i} \sim \mathcal{D}$
7:             Draw $b_\ell$ samples from $\mathcal{D}$ but terminate early if $t$ occurrences of $\boldsymbol{i}$ are observed.
8:             $\boldsymbol{X} \leftarrow$ number of samples drawn
9:             $\hat{\boldsymbol{H}}_\ell \leftarrow \hat{\boldsymbol{H}}_\ell + \log(\boldsymbol{X}/t) \cdot \mathbb{1}\{\boldsymbol{X} \in I_\ell\}$ and $\boldsymbol{c}_\ell \leftarrow \boldsymbol{c}_\ell + \mathbb{1}\{\boldsymbol{X} \in I_\ell\}$
10:        Define

$$\hat{\boldsymbol{q}}_\ell \leftarrow \begin{cases} c_\ell/r_\ell & \text{if } \ell < L \\ 1 - \sum_{\ell=1}^{L-1} \hat{\boldsymbol{q}}_\ell & \text{if } \ell = L \end{cases} \quad \text{and} \quad \hat{\boldsymbol{H}}_\ell \leftarrow \begin{cases} \hat{\boldsymbol{H}}_\ell/c_\ell & \text{if } c_\ell > 0 \\ H_\ell & \text{if } c_\ell = 0, \ell < L \\ \log(b_L/t) & \text{if } c_\ell = 0, \ell = L \end{cases}$$

        $\triangleright$ Note that if $c_\ell = 0$ and $\ell < L$, the definition of $\hat{\boldsymbol{H}}_\ell$ is just for analysis since in that case $\hat{\boldsymbol{q}}_\ell = 0$ and the output does not depend on $\hat{\boldsymbol{H}}_\ell$.
11:        $\hat{\boldsymbol{H}} \leftarrow \hat{\boldsymbol{H}} + \hat{\boldsymbol{q}}_\ell \hat{\boldsymbol{H}}_\ell$

---

**Lemma 3.3.** *Define* Error $:= \sum_{\ell=1}^L \hat{\boldsymbol{q}}_\ell \cdot \hat{\boldsymbol{H}}_\ell - \sum_{\ell=1}^L q_\ell \cdot H_\ell$. *For any setting of* $t = b_0 < \cdots < b_L = X_{\max}$ *and* $\{r_\ell \in \mathbb{N} : \ell \in [L]\}$, *we have*

$$\mathbf{E}\big[\mathsf{Error}^2\big] \leq 2 \cdot \sum_{\ell=1}^{L-1} \frac{q_\ell \log^2(b_L/b_{\ell-1})}{r_\ell} + 6 \cdot \sum_{\ell=1}^L \frac{q_\ell \log^2(b_\ell/b_{\ell-1})}{r_\ell + 1}$$

*Proof of Lemma 3.3.* We start by rewriting Error as follows:

$$
\begin{aligned}
\mathsf{Error} &= \sum_{\ell=1}^L (\hat{\boldsymbol{q}}_\ell - q_\ell)\,\hat{\boldsymbol{H}}_\ell + \sum_{\ell=1}^L q_\ell \left(\hat{\boldsymbol{H}}_\ell - H_\ell\right) \\
&= \sum_{\ell=1}^{L-1} (\hat{\boldsymbol{q}}_\ell - q_\ell)\,\hat{\boldsymbol{H}}_\ell + \left(1 - \sum_{\ell=1}^{L-1}\hat{\boldsymbol{q}}_\ell - 1 + \sum_{\ell=1}^{L-1} q_\ell\right)\hat{\boldsymbol{H}}_L + \sum_{\ell=1}^L q_\ell\left(\hat{\boldsymbol{H}}_\ell - H_\ell\right) \\
&= \sum_{\ell=1}^{L-1} (\hat{\boldsymbol{q}}_\ell - q_\ell)\left(\hat{\boldsymbol{H}}_\ell - \hat{\boldsymbol{H}}_L\right) + \sum_{\ell=1}^L q_\ell\left(\hat{\boldsymbol{H}}_\ell - H_\ell\right). \quad (5)
\end{aligned}
$$

Using the fact that $\hat{\boldsymbol{q}}_\ell$ is an unbiased estimator for $q_\ell$ and that for $\ell \leq L-1$, $\hat{\boldsymbol{H}}_\ell$ is an unbiased estimator of $H_\ell$ even when conditioned on a specific value of $\hat{\boldsymbol{q}}_\ell$, we have

$$\forall \ell \neq \ell' \in [L-1] \; : \; \mathbf{E}\Big[(\hat{\boldsymbol{q}}_\ell - q_\ell)\left(\hat{\boldsymbol{H}}_\ell - \hat{\boldsymbol{H}}_L\right)(\hat{\boldsymbol{q}}_{\ell'} - q_{\ell'})\left(\hat{\boldsymbol{H}}_{\ell'} - \hat{\boldsymbol{H}}_L\right)\Big] = 0$$

and

$$\forall \ell \neq \ell' \in [L] \; : \; \mathbf{E}\Big[q_\ell\left(\hat{\boldsymbol{H}}_\ell - H_\ell\right)q_{\ell'}\left(\hat{\boldsymbol{H}}_{\ell'} - H_{\ell'}\right)\Big] = 0 \,.$$

Hence,

$$
\begin{aligned}
\mathbf{E}\big[\mathsf{Error}^2\big] &\leq 2 \cdot \mathbf{E}\left[\left(\sum_{\ell=1}^{L-1}(\hat{\boldsymbol{q}}_\ell - q_\ell)\left(\hat{\boldsymbol{H}}_\ell - \hat{\boldsymbol{H}}_L\right)\right)^2\right] + 2 \cdot \mathbf{E}\left[\left(\sum_{\ell=1}^L q_\ell\left(\hat{\boldsymbol{H}}_\ell - H_\ell\right)\right)^2\right] \\
&= 2 \cdot \mathbf{E}\left[\sum_{\ell=1}^{L-1}(\hat{\boldsymbol{q}}_\ell - q_\ell)^2\left(\hat{\boldsymbol{H}}_\ell - \hat{\boldsymbol{H}}_L\right)^2\right] + 2 \cdot \mathbf{E}\left[\sum_{\ell=1}^L q_\ell^2\left(\hat{\boldsymbol{H}}_\ell - H_\ell\right)^2\right] \\
&\leq 2 \cdot \sum_{\ell=1}^{L-1} \mathbf{Var}[\hat{\boldsymbol{q}}_\ell]\,(\log(b_L/b_{\ell-1}))^2 + 2 \cdot \sum_{\ell=1}^L q_\ell^2 \mathbf{E}\left[\left(\hat{\boldsymbol{H}}_\ell - H_\ell\right)^2\right] \quad (6)
\end{aligned}
$$

Recall $\hat{\boldsymbol{q}}_\ell$ is the average of $r_\ell$ Bernoulli random variables, each set to 1 with probability $q_\ell$. Hence,

$$\mathbf{Var}[\hat{\boldsymbol{q}}_\ell] = \frac{q_\ell(1-q_\ell)}{r_\ell} \leq \frac{q_\ell}{r_\ell+1} \ . \tag{7}$$

To bound the expectation of $(\hat{\boldsymbol{H}}_\ell - H_\ell)^2$ recall the definition of $\hat{\boldsymbol{H}}_\ell$: we take a sample $\boldsymbol{i} \sim \mathcal{D}$, then $\boldsymbol{X} \sim \mathrm{NB}(t, p_{\boldsymbol{i}})$ to check whether $\boldsymbol{X}' \in I_\ell$; if so, we increment $\boldsymbol{c}_\ell$ and use $\log(\boldsymbol{X}'/t)$ as an estimate for $H_\ell$. Crucially, if we condition on any non-zero value of $\boldsymbol{c}_\ell$, $\hat{\boldsymbol{H}}_\ell$ is an unbiased estimator given by averaging $\boldsymbol{c}_\ell$ values, all of which are bounded in the interval $[\log(b_{\ell-1}/t), \log(b_\ell/t)]$. If $\boldsymbol{c}_\ell = 0$, the estimator sets $\hat{\boldsymbol{H}}_\ell$ to $\log(b_\ell/t)$. In particular, since for all non-zero positive integers, $1/\boldsymbol{c} \leq 2/(\boldsymbol{c}+1)$, we may write

$$\mathbf{E}\left[\left(\hat{\boldsymbol{H}}_\ell - H_\ell\right)^2\right] \leq \mathbf{Pr}[\boldsymbol{c}_\ell = 0] \cdot \log^2(b_\ell/b_{\ell-1}) + 2\mathbf{E}_{\boldsymbol{c}_\ell}\left[\frac{(\log(b_\ell/t) - \log(b_{\ell-1}/t)^2}{\boldsymbol{c}_\ell + 1}\right]$$

$$\leq q_\ell(1-q_\ell)^{r_\ell} \cdot \log^2(b_\ell/b_{\ell-1}) + 2\mathbf{E}_{\boldsymbol{c}_\ell}\left[\frac{1}{\boldsymbol{c}_\ell+1}\right]\log^2(b_\ell/b_{\ell-1})$$

$$\leq \frac{1}{1+r_\ell} \cdot \log^2(b_\ell/b_{\ell-1}) + \frac{2}{q_\ell(1+r_\ell)}\log^2(b_\ell/b_{\ell-1})$$

$$\leq \frac{3}{q_\ell(r_\ell+1)}\log^2(b_\ell/b_{\ell-1}) \tag{8}$$

where the second last inequality used that $q(1-q)^r \leq 1/(r+1)$ for any $q \in [0,1]$ and that since $\boldsymbol{c}_\ell$ is distributed as $\mathsf{Bin}(q_\ell, r_\ell)$, we have $\mathbf{E}_{\boldsymbol{c}_\ell}\left[(\boldsymbol{c}_\ell+1)^{-1}\right] \leq 1/(q_\ell(r_\ell+1)$. The proof of this last inequality appears as Lemma 6 in [ABIS19], which we reproduce below for convenience:

$$\mathbf{E}_{\boldsymbol{c}\sim\mathsf{Bin}(q,r)}\left[\frac{1}{1+\boldsymbol{c}}\right] = \sum_{l=0}^{r}\binom{r}{l}q^l(1-q)^{r-l}\cdot\frac{1}{l+1} = \sum_{l=0}^{r}\binom{r}{l}q^l(1-q)^{r-l}\cdot\frac{1}{r+1}\cdot\frac{r+1}{l+1}$$

$$= \frac{1-(1-q)^r}{q(r+1)} \leq \frac{1}{q(r+1)}.$$

Substituting Eq. 7 and Eq. 8 into Eq. 6 gives the lemma. □

We consider the following setting of parameters, where we let $L = \log^* k$,[8] such that we have $b_0 = t$, and

$$\forall \ell \in \{1, \ldots, L-1\}, \quad b_\ell \overset{\text{def}}{=} \frac{tk}{(\log^{(\ell)} k)^3} \quad , \quad b_L \overset{\text{def}}{=} \frac{tk}{\epsilon} \quad \text{and let} \quad r_\ell \overset{\text{def}}{=} \frac{80\log^2(b_L/b_{\ell-1})}{\epsilon^2}. \tag{9}$$

It is fairly straightforward to show that Algorithm 1 uses a constant number of words. Note that $r_\ell$ (similarly $b_\ell$'s) can be computed from $r_{\ell-1}$, so we do not need to calculate and store all the $r_\ell$'s beforehand. Also, to compute $\hat{\boldsymbol{q}}_L$, we do not need all the $\hat{\boldsymbol{q}}_\ell$'s. We only need to keep a running sum of $\hat{\boldsymbol{q}}_\ell$. Thus, we only need a constant number of words of memory. We analyze correctness simply by bounding the variance. Namely, the remainder of the section will be devoted to proving the following lemma, which will imply that our estimator will be within $\pm\epsilon$ of $\tilde{H}$ with constant probability.

**Lemma 3.4.** *The expected sample complexity of the algorithm is $O(k\log^4(1/\epsilon)/\epsilon^2)$ and returns an $\pm O(\epsilon)$ approximation with probability* 0.9.

## 4  Conclusions

We presented an algorithm for returning an additive $\epsilon$ approximation of the Shannon entropy of a distribution over $[k]$. The algorithm required $O(k\,\epsilon^{-2}\log^4(1/\epsilon))$ i.i.d. samples from the unknown distribution and a constant number of words of memory. In terms of the $\epsilon$ dependence, this improves over the state-of-the-art [ABIS19] by a factor $1/\epsilon$ in the sample complexity. More generally, we expect that the technique used, that of correcting the bias via low-degree polynomials will be useful in the context of the other inference problems in the data stream setting. The main open problem is determining whether the sample complexity of our result is optimal. We conjecture that this is the case, up to the $\mathrm{poly}(\log(1/\epsilon))$ factors. See Section D for a discussion of this conjecture.

---

[8]Recall that $\log^* z$ is the number of iterated logarithms (base 2) before the result is less than or equal to 1.

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
