## A Variance of `LogEstimator`

We now bound the variance of our estimator by $O(\log^2 k)$. Recall that the output of `LogEstimator` is given by $\log(\mathbf{X}/t) - g(\mathbf{B}_1, \ldots, \mathbf{B}_r)$, where the function $g$ is bounded. Since the variance we seek is $O(\log^2 k)$, it suffices to show that the variance of $\log(\mathbf{X}/t)$ is $O(\log^2 k)$ with $i \sim \mathcal{D}$, since subtracting $g$ changes the estimate by at most a constant (see Lemma 2.3).

**Lemma A.1.** *Let $i \sim \mathcal{D}$ and $\mathbf{X}$ denote the number of independent trials from $\mathrm{Ber}(p_i)$ before we see $t$ successes. Then, $\mathbf{Var}[\log(\mathbf{X}/t)] = O(\log^2 k)$.*

*Proof.* Let $X_{\max} = 2kt$, and consider the random variable $\mathbf{X}' = \min\{\mathbf{X}, X_{\max}\}$. Then

$$
\begin{aligned}
\mathbf{Var}[\log(\mathbf{X}/t)] &\leq \mathbf{E}\Big[\big(\log(\mathbf{X}/t) - \log(\mathbf{X}'/t) + \log(\mathbf{X}'/t)\big)^2\Big] \\
&\leq 2 \cdot \mathbf{E}\Big[\big(\log(\mathbf{X}/t) - \log(\mathbf{X}'/t)\big)^2\Big] + 2 \cdot \mathbf{E}\big[\log^2(\mathbf{X}'/t)\big] \\
&\leq 2 \cdot \mathbf{E}\big[\log^2(\mathbf{X}/\mathbf{X}')\big] + 2\log^2(2k) \\
&\leq \frac{4}{\ln^2(2)} \cdot \mathbf{E}\left[\left(\sqrt{\frac{\mathbf{X}}{\mathbf{X}'}} - 1\right)^2\right] + 2\log^2(2k),
\end{aligned}
$$

where we used that $\log(\mathbf{X}'/t) \leq \log(2k)$ always, and that $\log(z) \leq \sqrt{z-1}/\ln(2)$ for all $z \geq 1$. Then,

$$
\mathbf{E}\left[\frac{\mathbf{X}}{\mathbf{X}'} - 1\right] \leq \mathbf{E}\left[\frac{\mathbf{X}}{X_{\max}}\right] = \frac{1}{X_{\max}}\sum_{i=1}^{k} p_i \cdot \frac{t}{p_i} = \frac{tk}{X_{\max}} = 2.
$$

$\square$

## B Omitted Details from Section 2

*Proof of Claim 2.5.* Notice that $\mathbf{X}$ is the number of trials from $\mathrm{Ber}(p_i)$ until we see $t$ successes. We now have the following string of equalities:

$$
\begin{aligned}
\mathbf{E}_{\mathbf{X},\mathbf{B}_1,\ldots,\mathbf{B}_r}\left[\boldsymbol{\eta} - \log\left(\frac{1}{p_i}\right)\right] &= \mathbf{E}_{\mathbf{X}}[\log \mathbf{Y}] - \mathbf{E}_{\mathbf{B}_1,\ldots,\mathbf{B}_r}[g(\mathbf{B}_1, \mathbf{B}_2, \ldots, \mathbf{B}_r)] \\
&= \mathbf{E}_{\mathbf{X}}[f(\mathbf{Y}) + h(\mathbf{Y})] - g(p_i, p_i^2, \ldots, p_i^r) = \mathbf{E}_{\mathbf{X}}[h(\mathbf{Y})],
\end{aligned}
$$

where we used the fact that $g$ is a linear function, and that $\mathbf{E}[\mathbf{B}_\ell] = p_i^\ell$ in order to substitute

$$
\mathbf{E}_{\mathbf{B}_1,\ldots,\mathbf{B}_r}[g(\mathbf{B}_1, \ldots, \mathbf{B}_r)] = g(p_i, p_i^2, \ldots, p_i^r).
$$

Furthermore, we divide $\log \mathbf{Y} = f(\mathbf{Y}) + h(\mathbf{Y})$, where $f(z)$ is the degree-$r$ Taylor expansion of $\log z$ at 1, and $h(z) = \log z - f(z)$ is the error in the degree-$r$ Taylor expansion of $\log(z)$, i.e.,

$$
h(z) = \log(z) - f(z).
$$

Finally, by construction of $g$, $\mathbf{E}[f(\mathbf{Y})] = g(p_i, p_i^2, \ldots, p_i^r)$, which gives the desired equality. $\square$

**Verifying $\mathbf{Y}$ is subgamma.** Recall that $\mathbf{X}$ is the number of independent draws from a $\mathrm{Ber}(p)$ distribution until we see $t$ successes. In other words, we may express $\mathbf{X} = \mathbf{X}_1 + \cdots + \mathbf{X}_t$, where $\mathbf{X}_i$ is the number of draws of $\mathrm{Ber}(p)$ before we get a single success. Then, we always satisfy

$$
\mathbf{E}[\mathbf{X}_i] = \frac{1}{p} \qquad \mathbf{Pr}[\mathbf{X}_i > \ell] = (1-p)^{\lceil \ell \rceil} < e^{-p\ell}.
$$

This, in turn, implies that for any $r \geq 1$

$$
\left(\mathbf{E}[|\mathbf{X}_i - 1/p|^r]\right)^{1/r} \leq \left(\mathbf{E}_{\mathbf{X}_i, \mathbf{X}_i'}[|\mathbf{X}_i - \mathbf{X}_i'|^r]\right)^{1/r} \leq 2\left(\mathbf{E}[|\mathbf{X}_i|^r]\right)^{1/r} = O(r/p),
$$

where the first line is by Jensen's inequality, and the second is by the triangle inequality and Hölder inequality. Finally, we use the tail bound on $\boldsymbol{X}_i$ to upper bound the expectation of $|\boldsymbol{X}_i|^r$. Then, we have

$$
\mathbf{E}\left[e^{\lambda(\boldsymbol{X}_i - 1/p)}\right] = 1 + \lambda\mathbf{E}[\boldsymbol{X}_i - 1/p] + \sum_{k=2}^{\infty}\frac{\lambda^k}{k!}\cdot\mathbf{E}[|\boldsymbol{X}_i - 1/p|]
$$

$$
= 1 + \sum_{k=2}^{\infty}\frac{\lambda^k}{k!}\left(O(k/p)\right)^k \leq 1 + O(\lambda^2/p^2), \qquad \text{when } |\lambda| \text{ sufficiently smaller than } p
$$

$$
\leq \exp\left(O(\lambda^2/p^2)\right)
$$

Then, since $\boldsymbol{X}_1, \ldots, \boldsymbol{X}_t$ are all independent, we have

$$
\mathbf{E}\left[e^{\lambda(\boldsymbol{X} - t/p)}\right] \leq \exp\left(O(\lambda^2 t/p^2)\right) \implies \mathbf{E}\left[e^{\lambda(\boldsymbol{Y} - 1)}\right] \leq \exp\left(O(\lambda^2/t)\right),
$$

and this bound is valid whenever $|\lambda|$ is sufficiently smaller than $t$.

## C   Omitted Proofs from Section 3

*Proof of Lemma 3.1.* The approach is to estimate

$$
\mathbf{E}_{\boldsymbol{i}\sim\mathcal{D}}[h_t(p_{\boldsymbol{i}})] = \mathbf{E}_{\boldsymbol{i}\sim\mathcal{D}}\left[g(p_{\boldsymbol{i}}, p_{\boldsymbol{i}}^2, \ldots, p_{\boldsymbol{i}}^r)\right]. \tag{10}
$$

There exists an algorithm using $O(\log(1/\epsilon)/\epsilon^2)$ samples to estimate the above quantity: for $j \in \{0, \ldots, O(1/\epsilon^2)\}$, one takes a sample $\boldsymbol{i}_j \sim \mathcal{D}$ and uses $r = O(\log(1/\epsilon))$ additional samples $\boldsymbol{s}_1, \ldots, \boldsymbol{s}_r \sim \mathcal{D}$ to define

$$
\boldsymbol{B}_m^{(j)} \stackrel{\text{def}}{=} \mathbb{1}\{\boldsymbol{s}_1 = \cdots = \boldsymbol{s}_m = \boldsymbol{i}_j\} \quad \text{and} \quad \boldsymbol{Z}_j = g(\boldsymbol{B}_1^{(j)}, \ldots, \boldsymbol{B}_r^{(j)}).
$$

Then, let $\boldsymbol{Z}$ be the average of all $\boldsymbol{Z}_j$'s, which is an unbiased estimate to $\mathbf{E}_{\boldsymbol{i}\sim\mathcal{D}}\left[g(p_{\boldsymbol{i}}, p_{\boldsymbol{i}}^2, \ldots, p_{\boldsymbol{i}}^r)\right]$. Since $g$ is bounded (from Lemma 2.3), the variance of $O(1/\epsilon^2)$ such values is a large constant factor smaller than $\epsilon^2$. By Chebyshev's inequality, we estimate (10) to error $\pm\epsilon$ with probability at least 0.9. With that estimate, we will now use Lemma 2.4. Specifically, the entropy of $\mathcal{D}$ is exactly $\mathbf{E}_{\boldsymbol{i}\sim\mathcal{D}}[\log(1/p_{\boldsymbol{i}})]$, and we have

$$
\left|\mathbf{E}_{\boldsymbol{i}\sim\mathcal{D}}[\log(1/p_{\boldsymbol{i}})] - \left(\hat{H} - \boldsymbol{Z}\right)\right| \leq \epsilon + \left|\mathbf{E}_{\boldsymbol{i}\sim\mathcal{D}}[\log(1/p_{\boldsymbol{i}})] - \left(\hat{H} - \boldsymbol{Z}\right)\right|
$$

$$
\leq \epsilon + \mathbf{E}_{\boldsymbol{i}\sim\mathcal{D}}\left[\left|\log\left(\frac{1}{p_{\boldsymbol{i}}}\right) - \mathbf{E}[\boldsymbol{\eta}_{\boldsymbol{i}}]\right|\right] \leq 2\epsilon,
$$

where $\boldsymbol{\eta}_{\boldsymbol{i}}$ is the result of running `LogEstimator`$(\mathcal{D}, \boldsymbol{i})$. $\qquad\square$

*Proof of Lemma 3.2.* We note that since $\log(\cdot)$ is monotone increasing, we must have $H \geq \tilde{H}$. To see that it is not much larger, note that we always have $\log z = \ln(z)/\ln(2) \leq (z-1)/\ln(2)$, which means

$$
H - \tilde{H} = \mathbf{E}_{\boldsymbol{i},\boldsymbol{X}}\left[\log(\boldsymbol{X}/\boldsymbol{X}')\right] \leq \frac{1}{\ln(2)}\mathbf{E}_{\boldsymbol{i},\boldsymbol{X}}\left[\frac{\boldsymbol{X}}{\min\{\boldsymbol{X}, \boldsymbol{X}_{\max}\}} - 1\right] \leq \frac{1}{\ln(2)}\mathbf{E}_{\boldsymbol{i},\boldsymbol{X}}\left[\frac{\boldsymbol{X}}{\boldsymbol{X}_{\max}}\right]
$$

$$
= \frac{1}{\boldsymbol{X}_{\max}\cdot\ln(2)}\sum_{i=1}^{k}p_i\cdot\frac{t}{p_i} = \frac{tk}{\boldsymbol{X}_{\max}\cdot\ln(2)} = \epsilon.
$$

$\qquad\square$

*Proof of Lemma 3.4.* Substituting the $r_\ell$ values into Lemma 3.3 ensures $\mathbf{E}\left[\text{Error}^2\right] \leq \epsilon^2/10$. Hence the estimator is within $\pm\epsilon$ of $\tilde{H}$ with probability 0.9 by Chebyshev's inequality.

For the intervals $\ell = \{1, \ldots, L-1\}$, we always spend $r_\ell$ tries to determine whether a sample falls within a particular interval. Note that we take one sample to determine $\boldsymbol{i} \sim \mathcal{D}$, and then we take at

most $b_\ell$ samples. Therefore, the sample complexity for these is

$$\sum_{\ell=1}^{L-1} r_\ell \cdot b_\ell = \frac{80tk}{\epsilon^2} \cdot \sum_{\ell=1}^{L-1} \frac{\log^2(\log^{(\ell-1)}(k)/\epsilon)}{(\log^{(\ell)} k)^3} = \frac{80tk}{\epsilon^2} \cdot \sum_{\ell=1}^{L-1} \frac{(3\log^{(\ell)}(k) + \log(1/\epsilon))^2}{(\log^{(\ell)} k)^3}$$
$$\leq kt \cdot O(\log^2(1/\epsilon)/\epsilon^2),$$

where we used the fact that

$$\sum_{\ell=1}^{L-1} \frac{1}{(\log^{(\ell)} k)} \leq \frac{1}{1} + \frac{1}{\exp(1)} + \frac{1}{\exp(\exp(1))} + \frac{1}{\exp(\exp(\exp(1)))} + \ldots = O(1) .$$

Finally, it remains to bound the expected sample complexity of the bucket $L$. Here, we note

$$r_L = \frac{O(1)}{\epsilon^2} \cdot \log^2\left(\frac{\log^{(L-1)} k}{\epsilon}\right) \leq O\left(\frac{\log^2(1/\epsilon)}{\epsilon^2}\right) .$$

Therefore, the expected sample complexity for interval $L$ is $r_L \cdot \sum_{i=1}^{k} p_i \cdot \frac{t}{p_i} = O(k \log^4(1/\epsilon)/\epsilon^2)$.

$\square$

# D   Conjectured Lower Bound

Recall that without a memory constraint the sample complexity is known to be $n = \Theta(\max\{\epsilon^{-1} \cdot k/\log(k/\epsilon), \epsilon^{-2} \log^2 k\})$ [VV17, VV11, JVHW15, WY16]. To prove a $\Omega(k/\epsilon^2)$ lower bound for the memory constrained version, we conjecture the following randomized process can be used to generate distributions over $[2k]$ that look alike to any constant space algorithm that uses $o(k/\epsilon^2)$ samples but they have *different* entropies.

Suppose we have $k$ Bernoulli random variables with parameter $\alpha$: $Y_1, \ldots, Y_k$. And, we have $k$ Rademacher random variables $Z_1, \ldots, Z_k$ (that are $+1$ or $-1$ with probability $1/2$). We construct distribution $p$ in such a way that it is uniform over $k$ pairs of elements $(1, 2), (3, 4), \ldots, (2k - 1, 2k)$. However, conditioning on pair $(2i - 1, 2i)$, we may have a constant bias based on the random variable $Y_i$. And, we decide about the direction of the bias based on $Z_i$. More precisely, we set the probabilities in $p$ as follows:

$$p_{2i-1} = \frac{1 + Y_i \cdot Z_i/4}{2k} , \qquad p_{2i} = \frac{1 - Y_i \cdot Z_i/4}{2k} \qquad \forall i \in [k] .$$

Now, it is not hard to show that if we generate two distributions as above with $\alpha = (1 + \epsilon)/2$ and $\alpha = (1 - \epsilon)/2$, then their entropies are $\Theta(\epsilon)$ separated with a constant probability. Thus, any algorithm that can estimate the entropy has to *distinguish* $\alpha = (1 + \epsilon)/2$ from $\alpha = (1 - \epsilon)/2$. Intuitively, to learn $\alpha$, we would require to *determine* $\Omega(1/\epsilon^2)$ many of $Y_i$'s. Since we have only a constant words of memory, we cannot perform the estimation of the $Y_i$'s in parallels. Thus, any natural algorithm would require to draw $\Omega(k/\epsilon^2)$ samples.