# OpenReview forum: "Estimation of Entropy in Constant Space with Improved Sample Complexity"
_NeurIPS.cc/2022/Conference — NeurIPS 2022 Accept_

### Official Review · Reviewer_43yP · 2022-07-09

**Rating:** 6
**Confidence:** 5
**Soundness:** 4 excellent
**Presentation:** 3 good
**Contribution:** 3 good

**Summary:**

This paper investigates the entropy estimation problem under O(1) words of memory restriction, improving the current state-of-the art sampling complexity O(k polylog(1/ε)/ε^3) by a factor of 1/ε, where k is the alphabet size. The proposed algorithm inherits the framework of previous works, i.e. firstly take a random sample id, and use the number of samples equal to i to estimate p_i. Instead of sampling from binomial distributions, the new algorithm samples negative binomial variables to make the bias able to be estimated, and then adopt a Taylor expansion to correct the bias. Finally by incorporating a bucketing strategy, the reliance of sampling complexity on k is reduced, yielding a strict improvement compared with the previous work.

**Questions:**

The sample complexity is only bounded in expectation due to the property of negative binomial, which implies that the practical complexity may severely exceed the expected one with small probability.

**Limitations:**

1. The authors assume that r is far less than sqrt(t) in line 148. But they then states that r = O(log(1/ε)) and t = O(log^2(1/ε)) in line 149, which imply r ≈ sqrt(t). It seems inconsistent with the previous assumption.
2. The proof of Lemma 2.6 seems flawed, since when z > 2 the Taylor expansion of log(z) centered at 1 does not even converge.
3. Some other polynomial expansions e.g. Chebyshev series, are proven to outperform Taylor expansion. Is it possible to further improve the complexity by using Chebyshev series?
4. The organization of the paper could be further improved.  Some conclusions (e.g. eq (2), line 177) are referenced before their first appearance (e.g. eq (2), line 144).


**Strengths And Weaknesses:**

1. The idea of using negative binomial variables instead to improve sampling complexity is interesting, and to my knowledge it is original.
2. The idea of using Taylor expansion to estimate the bias is also impressive. This is the main contribution of this paper, which substantially improves the sampling complexity.
3. The claims of the paper are all supported by theoretical results, which is comprehensive and convincing.

---

> ### Author Response · Authors · 2022-08-02
> **Response to Reviewer 43yP**
>
> We thank the reviewer for their feedback.
>
> Question: If a worst-case sample complexity bound is required, we could use the fact any algorithm that uses $\mu$ samples in expectation into one with $c\mu$ worst case sample complexity by terminating if the number of samples used exceeds $c\mu$ for some constant $c>1$. This just increases the failure probability by $1/c$ by Markov. However, for our algorithm the sample complexity will be even more tightly concentrated around the expectation than Markov guarantees since negative binomial distributions have exponential tail bounds.
>
> Limitation (1): We added clarification, since we realize our usage of the notation $\ll$ was confusing. In the proof, we require that $r$ is ``far less than'' $\sqrt{t}$ in terms of constant factors, i.e., the constants in the asymptotic notation are set so  $r / \sqrt{t}$ is an arbitrarily small constant, which one can do with $r = O(\log(1/\epsilon))$ and $t = O(\log^2(1/\epsilon))$.
>
> Limitation (2): Agreed. While $z$ will end up concentrated around $1$, it could be larger than 2 and we can't use the Taylor expansion in that case. However, that case can be argued by the weak upper bound $|h(z)| = |\log(z) - f(z)| \leq |\log(z)| + |f(z)|$ without changing the statement of the lemma because this case has such small probability. We updated the pdf.
>
> Limitation (3): Note that we only use the first $O(\log (1/\epsilon))$ terms of the Taylor expansion, and the degree of this polynomial is $O(\log(1/\epsilon))$. Using the Chebyshev polynomials to approximate log function would provide a polynomial with a potentially lower degree. While this is an interesting idea to investigate, we speculate it would likely at most save $\textup{polylog}(1/\epsilon)$ factors.

---

### Official Review · Reviewer_kziV · 2022-07-11

**Rating:** 7
**Confidence:** 4
**Soundness:** 4 excellent
**Presentation:** 4 excellent
**Contribution:** 4 excellent

**Summary:**

This paper considers the problem of estimating the entropy of a distribution D over a finite set [k], given access to a stream of i.i.d. samples from D. It is shown that k/eps^2*polylog(1/\eps) samples suffice for an algorithm with constant words of memory to estimate the entropy within an eps additive error. This improves the previous best sample complexity by a factor of 1/eps. The authors also conjectured that the bound is optimal for constant space algorithms, up to polylog(1/\eps) factors.

The main idea of the paper is to write H(D)=E[log(1/p_i)] where i is drawn from D, and to estimate log(1/p_i) for each i. The estimator used in this paper works by empirically estimating a negative binomial random variable X with mean t/p_i, which is the number of samples before seeing exactly t copies of i, and using log(X/t) as a reasonable estimate. The authors further observed that this estimator is biased (i.e., E[log(X/t)] is not equal to log(1/p_i)), and due to Taylor expansion the bias can be written as a polynomial in p_i plus a small remainder. Since the power p_i^r is equal to the probability that first r samples are all copies of i, the bias can thus be estimated to a high precision in constant space. To get rid of the polylog(k) factors, the authors further employed a bucketing technique, which comes down to estimating conditional expectations of log(X/t) given X belongs to some interval I, and the probabilities that X belongs to I.



**Questions:**

1. Expression (1) at the bottom of page 5 needs more explanation. The first inequality is non-trivial since there are negative terms in the summation. The last inequality, which I believe should be true, can fit in some extra steps of calculation. In fact, I think Taylor series with remainders can probably simplify the proof here (although I didn't check).
2. The discussion about space complexity before Lemma 3.3 does not make sense until the parameters (b_l, r_l, etc.) are defined.
3. What is an intuitive explanation of why the bucketing technique can remove polylog(k) factors?


**Strengths And Weaknesses:**

Strength: The main improving idea is simple yet novel and non-trivial. The previous best bound is achieved by estimating p_i in a straightforward way and arguing that the bias E[log(1/^p_i)]-E[log(1/p_i)] is small. This paper takes a different approach by identifying a relevant negative binomial random variable from the sample stream. Interestingly, the bias of this new estimator is a polynomial in p_i, which is easily approximated. The overall flow of the paper is smooth.

Weaknesses: It would be nice if the paper provides some applications of entropy estimation, especially in the context of streaming with constant memory constraint. Some places require more explanations. See below for specific questions.

---

> ### Author Response · Authors · 2022-08-02
> **Response to Reviewer kziV**
>
> We thank the reviewer for their feedback.
>
> Question (1): Yes, the negative terms in the sum make that inequality non-trivial and we have added more detail (it can be proved via pairing successive terms). Note we also had to add a different argument for when z≥2 because in that case the Taylor series does not converge.
>
> Question (2): Agreed. We moved that paragraph slightly later.
>
> Question (3): The motivation for writing ${\bf E}[\log(X/t)]$ as $\sum_i {\bf E}[\log(X/t)|X\in B_i] \Pr[X\in B_i]$ for some intervals $B_1, B_2, \ldots$  is that ${\bf Var}[\log(X/t) | X\in B_i]$ can be made smaller than $O(\log^2 k)$ by insuring the ratio of limits of each $B_i$ is small. Combined with the fact we may ignore large values of $X$ when sampling conditioned on $X \in B_i$ we can  estimate each ${\bf E}[\log(X/t)|X\in B_i]$ with fewer samples because of the smaller variance.

---

### Official Review · Reviewer_LvWV · 2022-07-12

**Rating:** 4
**Confidence:** 3
**Soundness:** 3 good
**Presentation:** 3 good
**Contribution:** 2 fair

**Summary:**

This paper considers estimating the entropy of an unknown distribution over an alphabet of size $k$ from independent samples. The model they consider is the *sample streaming model* where independent samples are streamed from the source of a distribution. The resources of interest are (a) memory used  and (b) number of samples required. This model has gained some interest recently as it captures a realistic situation where the data from a source distribution might be streaming in high velocity and estimations has to be done with limited memory without storing all the data. From a model point-of-view, the task is close to designing streaming algorithms over randomized streams (as oppose to worst-case streams).

The paper that precedes, and most relevant to, the current work is the one due to Acharya, Bhadane, Indyk, and Sun (appeared in NeurIPS 2019) where they first consider entropy estimation problem in the sample streaming model.  They designed a O(1) word algorithm with sample complexity $O({k\over \varepsilon^3}\cdot {\rm polylog}(1/\varepsilon))$ that estimates the entropy of an unknown distribution over an alphabet of size $k$ within an additive error $\varepsilon$.  Here a word of memory is $O(\log k + \log {1\over \varepsilon})$ bits of memory.

The main contribution of the present paper is a new algorithm for the entropy estimation problem in the sample streaming model with $O(1)$ word memory and $O({k\over \varepsilon^2}\cdot {\rm polylog}(1/\varepsilon))$ samples. Thus the sample complexity is improved by ${1\over \varepsilon}$ factor while the memory remains $O(1)$. The proof follows the structure of ABIS19 paper. But there are some crucial differences including that their basic estimator is different.  The paper conjectures that this bound is optimal up to ${\rm polylog}({1\over \varepsilon})$ factors if the memory needs to be constant.

**Questions:**

 Without a lower bound result, the feel of the paper is incremental.  Is any lower bound better than the one for the unlimited memory model ($\Omega({k\over \varepsilon \log k} + {\log^2 k \over \varepsilon^2})$) known? I cannot gauge the difficulty of this problem: but any concrete step towards a better lower bound on sample complexity for entropy estimation while maintaining $O(1)$ memory will make the paper substantially stronger.

**Limitations:**

I do not see a discussion of the limitations and potential negative societal impact of the work in the paper. However, this may not be relevant for this paper which is theoretical in nature.

**Strengths And Weaknesses:**

*Strengths:* The model and the problem considered are significant. The paper makes a concrete contribution by improving the sample complexity from prior paper which appeared in NeurIPS. The techniques seem to be  fairly deep and cleaver.

*Weaknesses:* The paper seems incremental. There is just one result. While it is clean, the contribution may not be of  interest to a wider NeurIPS audience as it is mainly technical. The conceptual contribution is weak. The conjecture is very nice. The authors provide some discussion of how to prove a lower bound in the appendix, but there is no formal lower bound proof  given for substantiating the conjecture other than the upper bound.

---

> ### Author Response · Authors · 2022-08-02
> **Response to Reviewer LvWV**
>
> We thank the reviewer for their feedback. We agree that proving a lower bound is an important open question but given the challenges of proving related results (e.g., recent STOC and FOCS papers whose main results are lower bounds on the sample/space complexity of testing the bias of a coin from iid coin flips) we don't want to underestimate the difficulty of proving our conjectured lower bound. But we think our result on the upper bound is still an important (and technically challenging) step in resolving the complexity of a natural problem in the space/sample complexity setting. Note we have also added a paragraph at the end of introduction to discuss how results by Diakonikolas et al. are not sufficient to improve the known lower bounds.

---

### Official Review · Reviewer_JqRg · 2022-07-18

**Rating:** 7
**Confidence:** 5
**Soundness:** 4 excellent
**Presentation:** 4 excellent
**Contribution:** 3 good

**Summary:**

This paper considers entropy estimation of an unknown discrete distribution of alphabet size $k$ under memory constraints, specifically constant number of "words". Previous work proposed an algorithm whose sample complexity is $\tilde{O}(k/\epsilon^3)$. This work improves the sample complexity to $\tilde{O}(k/\epsilon^2)$.
The entropy of a discrete distribution can be interpreted as the expectation of $\log(1/p(X))$. Therefore, the crux of the algorithm is in averaging over multiple instances an estimate of $\log(1/p(X))$. The key innovation in this work is a low-bias, low memory algorithm to estimate $\log(1/p(X))$. Instead of using a plug-in estimate as previous work, this paper samples a negative binomial random variable to see an element $i$ exactly $t$ times. Although this is an unbounded random variable, the crux of the improvement lies in noticing that in expectation, the number of samples is approximately $k$ for this step. Using a few more samples, the Taylor series approximation of $\log(1/p(X))$ can be computed and the bias is in turn the error of the Taylor series approximation. Instead of averaging these estimates, the paper considers a method of bucketing similar to previous work.

**Questions:**

Included in the previous section.

**Limitations:**

Perhaps the main limitation is lack of a matching lower bound. However, the authors have adequately addressed this difficulty in the appendix.

**Strengths And Weaknesses:**

Statistical estimation/learning under memory constraints has started recently receiving attention. While the few existing algorithms in the specific area of distribution learning/property estimation under memory constraints are relatively straightforward, their analysis is not. The paper provides an algorithm that improves the sample complexity of estimating entropy using constant number of words. The main idea behind the improvement is crisp. The presentation is quite clear and the paper is well written.

The lack of a matching lower bound is a weakness but the authors have adequately addressed that in the appendix. One suggestion is to cite the following paper (https://arxiv.org/pdf/1907.05816.pdf, Theorem 7) that provides an improved lower bound in the streaming model for additive entropy estimation, compared to CCM10.
Typo in line 246 - should be p_i instead of q_i?

---

> ### Author Response · Authors · 2022-08-02
> **Response to Reviewer JqRg**
>
> We thank the reviewer for their feedback. We added the suggested citation and thanks for catching the p vs. q typo. For further discussion on the lower bound please see our response to Reviewer LvWv.

---

> > ### Comment · Reviewer_JqRg · 2022-08-08
> > **Post-Rebuttal comment**
> >
> > Thank you for your response. I have no further clarifications.

---

### Meta-Review · Area_Chair_d1Fj · 2022-08-24

**Recommendation:** Accept
**Confidence:** Certain

**Metareview:**

This paper improves the state-of-the-art in estimating the entropy of a discrete distribution under memory constraints. The reviewers agreed that the presented result is elegant and non-trivial and that the ideas are described well. There is some concern that the paper is incremental due to the absence of lower bounds, but the algorithmic contribution is strong enough to merit acceptance to NeurIPS.

**Award:**

No

---

### Decision · Program_Chairs · 2022-09-14

Accept